# Land-use diversity predicts regional bird taxonomic and functional richness worldwide

Carlos Martínez-Núñez [1,2] ✉, Ricardo Martínez-Prentice [3] &
Vicente García-Navas [1,4]

Unveiling the processes that shape biodiversity patterns is a cornerstone of ecology. Land-use diversity (i.e., the variety of land-use categories within an area) is often considered an important environmental factor that promotes species richness at landscape and regional scales by increasing beta-diversity. Still, the role of land-use diversity in structuring global taxonomic and functional richness is unknown. Here, we examine the hypothesis that regional species taxonomic and functional richness is explained by global patterns of land-use diversity by analyzing distribution and trait data for all extant birds. We found strong support for our hypothesis. Land-use diversity predicted bird taxonomic and functional richness in almost all biogeographic realms, even after accounting for the effect of net primary productivity (i.e., a proxy of resource availability and habitat heterogeneity). This link was particularly consistent with functional richness compared to taxonomic richness. In the Palearctic and Afrotropic realms, a saturation effect was evident, suggesting a non-linear relationship between land-use diversity and biodiversity. Our results reveal that land-use diversity is a key environmental factor associated with several facets of bird regional diversity, widening our understanding of key large-scale predictors of biodiversity patterns. These results can contribute to policies aimed at minimizing regional biodiversity loss.

In the Anthropocene, the loss of land-use diversity (i.e., defined as variety of land-use categories within an area), is considered a main global change driver that contributes to biodiversity loss through biotic homogenization[1–3]. In contrast, regions with low land-use diversity but highly productive habitats (i.e., primary tropical forests) can support high amounts of biodiversity[4]. Understanding the general role of land-use diversity in structuring regional diversity across the globe is crucial to weighing the importance of large-scale environmental complexity beyond habitat quality (i.e., quality of a specific land-use category), and to assisting policies aimed at halting biodiversity loss at the regional scale. Yet, a universal direct link between land-use diversity and biodiversity patterns is still missing.

There is currently significant evidence that land-use change has pervasive effects on biodiversity[5,6]. However, it is still not clear how local changes scale up spatially to regional levels. The often-negative shifts induced by land-use change in local diversity might reverse when focusing on higher levels of complexity[7]. In fact, despite that more heterogeneous habitats (i.e., land-use types that show a high degree of structural complexity and often high productivity such as tropical forests) foster local and landscape biodiversity[8,9], promoting regional diversity might require

[1]Department of Integrative Ecology, Estación Biológica de Doñana EBD (CSIC), Seville, Spain. [2]Agroscope, Reckenholzstrasse 191, CH-8046 Zurich, Switzerland. [3]Institute of Agriculture and Environmental Sciences, Estonian University of Life Sciences, Tartu, Estonia. [4]Department of Evolutionary Biology and Environmental Studies, University of Zurich, Zurich, Switzerland. ✉e-mail: cmnunez@ujaen.es

land-use diversification[10,11] even if it implies mixing habitat types of a priori different quality[12]. For instance, increased heterogeneity due to shifts in land-use has been linked to positive effects on species richness at the landscape scale through increases in species beta-diversity[13]. This is due to the fact that the number of species supported by a specific habitat or land-use type saturates at a certain point[4,14]. Therefore, land-use diversity is expected to have a strong positive impact on regional species richness. However, despite this, other variables have historically received more attention as main global moderators of fauna community richness, namely: latitude[15–17], elevation[18–20], primary productivity[21,22] and associated climatic variables[23,24]. With human activity reducing land-use diversity in large areas (e.g., deforestation, urbanization or desertification, aggravated by the ongoing climate change)[25,26], uncovering the importance of this variable on biodiversity is key for conservation. Moreover, it is important to incorporate information about species functional traits, since these functional properties are a closer proxy of ecosystem functioning[17,27–29] and different facets of diversity might respond differently to environmental gradients[30,31].

Here, we used worldwide land-cover maps as well as information about the distribution and ecological traits of all extant bird species to test the hypothesis that regional land-use diversity increases bird taxonomic and functional richness across the six main biogeographic realms. According to ecological theory, different land-use types will support different species with contrasting characteristics[32,33], from what we expect that land-use diverse regions will imprint a strong positive signal on taxonomic and functional richness, regardless of habitat quality. Also, because new land-use types might host functionally different species, we predict that land-use diversity will influence functional richness more strongly than taxonomic richness. Finally, we also predict that the shape of the relationship between land-use diversity and taxonomic/functional species richness will be moderated by the degree of diversity and species specialization in each biogeographic realm, with this relationship being stronger in realms with more diverse and habitat-specialist rich communities.

In this work, we show that land-use diversity is an important predictor of regional bird taxonomic and functional richness worldwide.

## Results

From the total 10,649 unique bird species considered in this study (BirdLife International taxonomy as reference), 926 species appeared in the Nearctic realm (25% of them unique species to this realm), 1847 in the Palearctic realm (25% unique species), 2093 in the Indomalayan realm (44% unique species), 4228 species occurred in the Neotropic realm (84% unique species), 2214 in the Afrotropic realm (81% unique species) and 1892 in the Australasian realm (83% unique species).

Regional land-use diversity largely varied throughout the 15,780 grid cells considered (Fig. 1), with some minimum values corresponding to large, forested areas (e.g., Amazon rainforest) and deserts (e.g., Sahara, the Gobi). Maximum values were found in Canada, Central and Southern Europe, and Southwest America/Africa. Land-use diversity was not linked to any other major global driver of species richness (Supplementary Fig. 1), apart from net primary productivity (NPP) (Pearson: $r = 0.49$, $P < 0.001$). This positive relationship (with the exception of the Nearctic realm; Linear model slope = −0.234 [from −0.334 to −0.133, 95% CI]), was particularly strong in both the Palearctic (LM slope = 1.908 [from 1.863 to 1.953, 95% CI]) and the Afrotropic realms (LM slope = 1.044 [from 0.962 to 1.126, 95% CI]).

Land-use diversity was an important global factor explaining bird taxonomic richness ($\Delta$AIC = 765; $P < 0.001$; $\Delta R^2 = 0.12$) and bird functional richness ($\Delta$AIC = 3078; $P < 0.001$; $\Delta R^2 = 0.14$) (Supplementary Fig. 2). Land-use diversity was strongly linked to regional bird taxonomic and functional richness across the six main realms (Table 1; Fig. 2). The observed response of functional richness was not only explained by changes in taxonomic richness, since there was a positive link between land-use diversity and functional richness across realms even after controlling for the effect of taxonomic richness (Supplementary Fig. 3). Particularly, steep increases were detected in functional richness in the Palearctic, Neotropic, and Australasian realms, with a two- to three-fold increase compared to regions with lower land-use diversity. The increase in taxonomic richness with land-use diversity was partly driven by a parallel increase in NPP (Fig. 2; Supplementary Fig. 1). In fact, once the influence of NPP was removed, the impact of land-use diversity on taxonomic richness was importantly diminished (but still substantial). After accounting for NPP, effects remained strong on functional richness (Fig. 3) in all cases except the Nearctic realm. In the Nearctic, NPP was the most important factor (among the considered variables) influencing both taxonomic and

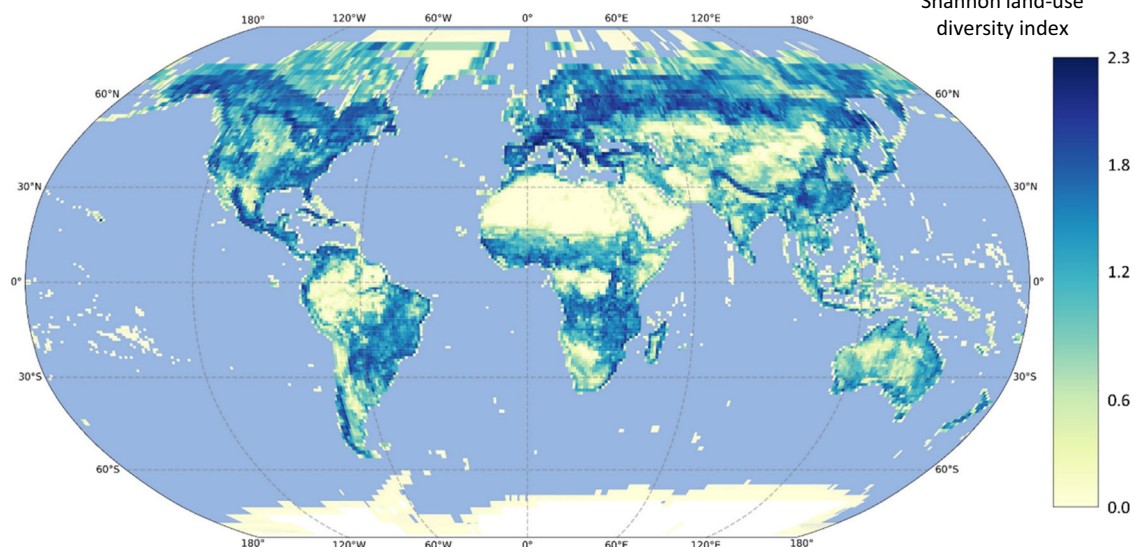

**Fig. 1 | Regional land-use diversity across the world.** Shannon land-use diversity index in each grid cell (one-degree size), calculated from Copernicus Land cover maps (~100 m resolution; 2019), which classifies land surface in 22 categories of land-use types. Robinson global projection. Base map made with Natural Earth.

**Table 1 | Competing models predicting global differences in bird taxonomic and functional richness**

| Model code | Response variable | Variables included | AIC | ΔAIC selected model | Deviance explained |
|---|---|---|---|---|---|
| **Mod_T1** | Taxonomic richness | Latitude + Longitude | −25,778 | 20,306 | 19.4% |
| **Mod_T2** | | Latitude + Longitude + NPP | −40,686 | 5398 | 67.7% |
| **Mod_T3** | | Latitude + Longitude + Land-use diversity | −37,072 | 9012 | 59.8% |
| **Mod_T4** | | Latitude + Longitude + NPP + Land-use diversity | −44,205 | 1879 | 74.3% |
| **Mod_T5** | | Latitude + Longitude + NPP + Land-use diversity + ME | −46,084 | 0 | 77.2% |
| **Mod_F1** | Functional richness | Latitude + Longitude | −6104 | 15,489 | 15.7% |
| **Mod_F2** | | Latitude + Longitude + NPP | −16,714 | 4879 | 55.4% |
| **Mod_F3** | | Latitude + Longitude + Land-use diversity | −15,563 | 6030 | 53.1% |
| **Mod_F4** | | Latitude + Longitude + NPP + Land-use diversity | −19,558 | 2035 | 62.9% |
| **Mod_F5** | | Latitude + Longitude + NPP + Land-use diversity + ME | −21,593 | 0 | 67.5% |

*NPP* Net primary productivity, *Land-use diversity* Shannon index of land-use diversity, *ME* Median elevation.

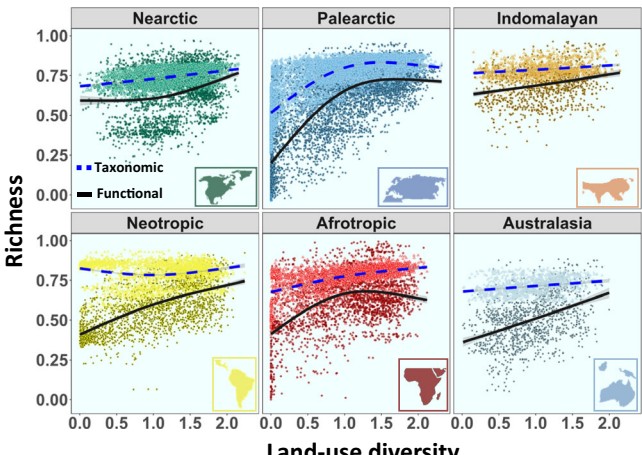

**Fig. 2 | Effect of regional land-use diversity on bird richness.** Additive model fit to partial residuals, showing the association between land-use diversity (Shannon index) and taxonomic richness (blue dashed line) and functional richness (black solid line) across the six main biogeographic realms. Each point represents a grid cell of one-degree size. Taxonomic and functional richness were standardized to the maximum value observed. Lines show the model fit and line shadows represent 95% confidence intervals. Nearctic: *n* = 2630 grid cells. Neotropic: *n* = 2262 grid cells. Palearctic: *n* = 6123 grid cells. Afrotropic: *n* = 2466 grid cells. Australasia: *n* = 1212 grid cells. Indomalaya: *n* = 1087 grid cells. Source data are provided as a Source Data file. Biogeographic realm outlines were modified from https://freesvg.org.

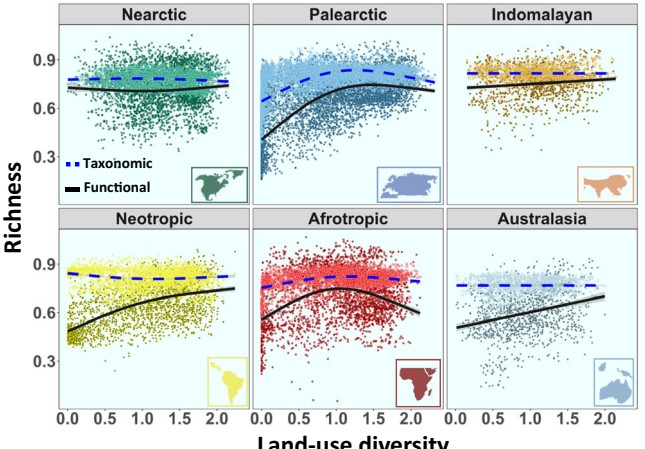

**Fig. 3 | Effect of regional land-use diversity on bird richness after accounting for primary productivity.** Additive model fit to partial residuals after accounting for primary productivity, showing the association between land-use diversity (Shannon index) and taxonomic richness (blue dashed line), and functional richness (black solid line) across the six main biogeographic realms. Each point represents a grid cell of one-degree size. Taxonomic and functional richness were standardized to the maximum value observed. Lines show the model fit and shadows represent 95% confidence intervals. Nearctic: *n* = 2630 grid cells. Neotropic: *n* = 2262 grid cells. Palearctic: *n* = 6123 grid cells. Afrotropic: *n* = 2466 grid cells. Australasia: *n* = 1212 grid cells. Indomalaya: *n* = 1087 grid cells. Source data are provided as a Source Data file. Biogeographic realm outlines were modified from https://freesvg.org.

functional diversity (Fig. 4; see Discussion). This realm supported a very low number of species compared to the other realms, most of these species also appeared in other realms, and it was the only realm in which the median size of the species range distribution increased with land-use diversity (Fig. 4).

Saturation effects of functional richness were observed in the Palearctic and Afrotropic realms at intermediate levels of land-use diversity (Fig. 2). This saturation turned into a hump-shaped pattern in the Afrotropic realm after controlling for taxonomic richness or primary productivity (Fig. 3).

## Discussion

Unveiling the processes that shape patterns of biodiversity is a key challenge in ecology. Land-use diversity (i.e., variability of land-use categories within an area) is often acknowledged as an important factor promoting biodiversity at large spatial scales, beyond smaller scales, where habitat heterogeneity (i.e., quality or heterogeneity of a specific land-use category) seems to be the main predictor[34]. However, the pervasive effects of fragmentation on the positive species-area

relationship might pose an early limit to the benefits of land-use diversification in real-world scenarios[35]. Here, we show, that regional land-use diversity constitutes a key factor strongly associated with bird taxonomic and functional diversity worldwide. This has important implications for our understanding of how environmental complexity at different scales affects biodiversity and can help the development of policies focused at larger spatial scales to mitigate regional biodiversity loss.

According to ecological theory[36], single land-use types can host a limited number of species with certain functional characteristics that enable them to thrive in that particular environment[37]. The richness of species a single land-use type can support depends on its heterogeneity and the number of niches it can provide, aspects often linked to net primary productivity[4,14]. In fact, our results show a generalized and strong positive association between NPP and bird taxonomic and functional richness, being NPP the most important predictor in the models (Table 1 and Supplementary Fig. 4). This supports the widely accepted idea that simplified environments such as arable lands or urban areas usually provide fewer niches, allowing a filtered or reduced

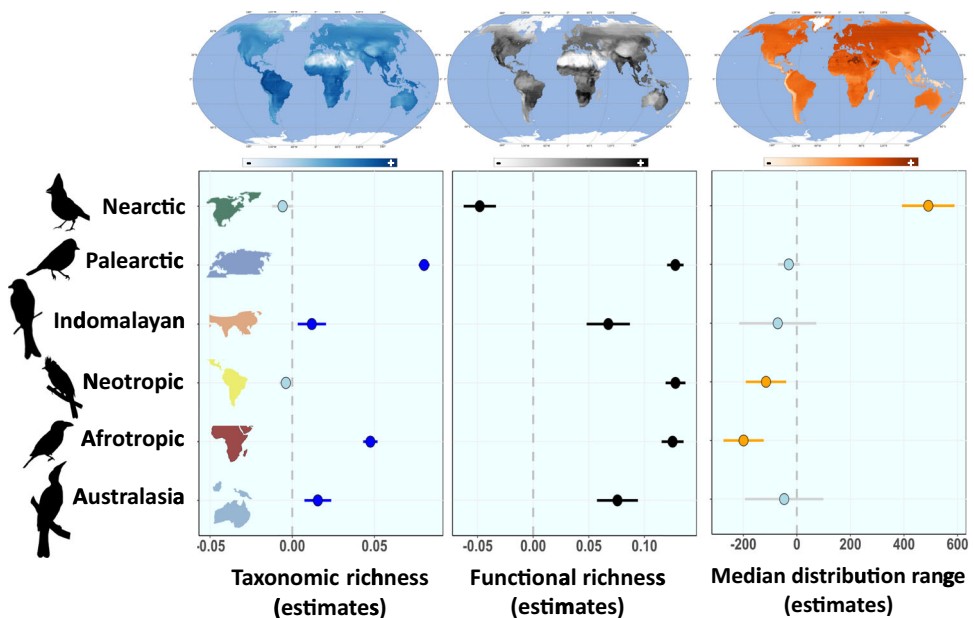

**Fig. 4 | Effect of land-use diversity on bird regional taxonomic, functional richness, and median community distribution range.** Circles represent estimated slopes (beta coefficient of linear trends ±95% CI) after accounting for the effect of primary productivity, for taxonomic richness (left panel, blue dots), functional richness (central panel, black dots) and median distribution range (right panel, orange dots). Non-significant trends (slope no different from 0) are shown in grey. Nearctic: $n = 2630$ grid cells. Neotropic: $n = 2262$ grid cells. Palearctic: $n = 6123$ grid cells. Afrotropic: $n = 2466$ grid cells. Australasia: $n = 1212$ grid cells. Indomalaya: $n = 1087$ grid cells. Source data are provided as a Source Data file. Biogeographic realm outlines were modified from https://freesvg.org. Birds silhouettes were downloaded from http://phylopic.org representing the following species: *Cardinalis cardinalis*, *Cyanistes caeruleus*, *Dicrurus leucophaeus*, *Myiarchus tyrannulus*, *Lybius torquatus*, and *Philemon corniculatus*.

set of species to survive in these land-use types[38–41]. However, our results show that, at a regional scale, land-use diversity might also imprint a strong signal on bird assemblages, and, apart from land-use or habitat quality, the diversity of land uses plays a central role in explaining bird taxonomic and, especially, functional richness worldwide. This strong and generalized association with land-use diversity suggests that diversity tends to saturate easily in real-world land-use types at this scale. After this saturation or deceleration in biodiversity gain with area, the increase in regional richness (gamma diversity) is mainly achieved through the addition of new land-use types that prominently increase beta-diversity (i.e., diversity between sites) by adding new whole sets of species from the regional pool of species[42,43]. The fact that functional richness increased more steeply than taxonomic richness supports this idea, since new land-use types, as defined by different land-covers here, might add functionally singular species, producing a disproportionately higher raise in the functional space per additional species unit added.

According to these findings, patches of highly heterogeneous land-use types (i.e., habitats) mixed at the regional scale should maximize diversity. However, there must be a trade-off between the size of these patches and the number of different land-use types that provides maximum gains, because habitat fragmentation and the lack of a specific habitat type can involve a high cost to many (mainly habitat specialists) species[44]. In fact, we found that richness was maximum at intermediate levels of land-use diversity in the Palearctic and the Afrotropic realms, probably due to these trade-offs. To maximize biodiversity at the regional scale, further studies should disentangle the mechanisms governing the trade-offs between land-use diversity and fragmentation, which are probably highly taxa-, scale- and context-dependent (i.e., depending on the fauna group, spatial scale, and habitat types involved). Additionally, even if fragmentation is inherently related to land-use diversity in terms of patch composition, a general global link between fragmentation and land-use diversity is still missing and should be identified in further studies.

The Nearctic was an exception to the general patterns found here. In this realm, NPP and land-use diversity were negatively correlated, with NPP being a more important predictor of taxonomic and functional richness. Three aspects might complementarily explain this rarity: (i) NPP might counteract and mask the positive effects of land-use diversity, since we find that NPP is a stronger predictor of taxonomic and functional richness and, in this realm, the NPP and land-use diversity oppose; (ii) this realm hosted only a small proportion of unique species, and, in addition, land-use diverse regions hosted species assemblages with overall bigger distribution range sizes (response only observed in this realm), which suggests that the occurrence of species with a low degree of specialization and a low preference for a specific habitat might reduce the level of community compartmentalization in different land-use types[33,45]; and (iii) the overall very low number of species observed compared to other realms, might also reduce the level of niche partitioning and delay richness saturation in single land-use types, reducing the benefits of land-use diversification. These three reasons might explain why we did not find an increase in richness with land-use diversity in the Nearctic. In addition, further studies should delve into the habitat-specific biodiversity-area saturation curve, to maximize the efficiency of regional land-use planning.

In conclusion, we show that regional bird taxonomic and functional richness are strongly associated with net primary productivity (i.e., a proxy of habitat heterogeneity and resource availability) and regional land-use diversity across the world. Further research should address if the generalized response found in birds is extensible to other fauna. These results have important implications for conservation policies because, of the factors that explain global patterns of species and functional richness, land-use diversity is perhaps one that can be most (and quickly) influenced by humans.

Moreover, human-induced impacts that promote land-use change globally[46], such as desertification[47], deforestation for agricultural expansion[48], or urbanization[49], often imply large-scale environmental

simplification, reducing land-use diversity and primary productivity simultaneously. These changes in the environment not only are detrimental for local biodiversity because they extend poor-quality land-use types (i.e., land-use types with low heterogeneity and net primary productivity), but also might lead to regional losses through biotic homogenization, two processes that synergistically endanger biodiversity[50] across the globe. Lastly, land-use diversity should also be taken into consideration to increase efficiency while designing protected areas. Further research should delve into how highly productive and land-use diverse regions overlap with important areas for birds.

## Methods

### Bird spatial data

We obtained spatially explicit bird distribution occurrence data from the AVONET database[51], which is based on data provided by BirdLife International (2019). Bird worldwide distributions were extracted to an equal area grid (Behrmann projection) with a resolution of ~1 degree (~110 km side cells). Only sites in which bird species were classified as breeding native, breeding reintroduced, or resident were selected. Because we were interested in understanding a fundamental pattern in species assemblages, we did not include introduced species, since their occurrence can be subject to more stochastic processes or specific events and could introduce some bias (e.g., some regions are more prone to host alien species due to different factors including tourism and commercial transactions). In total, 18,710 terrestrial grid cells covering the world were originally considered. Grid cells with a small proportion of land (<10%) or less than six different species were excluded, because at least six species were needed to characterize the functional space in each grid cell. Ultimately, 15,780 grid cells were used for analyses, belonging to the six main biogeographic realms: Nearctic (2630 grid cells), Neotropic (2262 grid cells), Palearctic (6123 grid cells), Afrotropic (2466 grid cells), Australasia (1212 grid cells), and Indomalaya (1087 grid cells).

### Taxonomic diversity, bird traits, and functional diversity

The taxonomic richness of bird species in each grid cell was calculated by counting the number of different species occurring in each grid cell. To calculate functional diversity, bird traits were obtained from the AVONET dataset, that assembles morphological, ecological, and geographic data about all extant bird species[51]. From this source, we considered several morphological and ecological traits that are functionally important (Supplementary Table 1). Trait selection aimed to account for as many important ecological facets of species' ecology as possible to characterize functional diversity in a comprehensive way, with the constrain of data availability for all the species in the world. We preselected: (i) body mass; (ii) four traits summarizing beak morphology: length from the tip to the culmen, length from the tip to the nares, beak width, and beak depth; (iii) four traits reflecting body morphology: length of tarsus, wing length, tail length, and the hand-wing index; (iv) diet or predominant trophic niche (categorical with ten levels; e.g., aquatic predator, insectivorous, granivorous, frugivorous, etc.); and (v) primary lifestyle (categorical with five levels, e.g., aerial, terrestrial, insessorial, etc.). We first ran two principal component analyses (PCA) to synthesize information about body morphology and beak traits, respectively. Traits were first log-transformed. The first PCA included the following traits: body mass, length of tarsus, wing length, and tail length. The first axis (PCA1$m$) was highly correlated with size (body mass), while the second one (PCA2$m$) was more related to shape. These two axes together accounted for 92% of the total variation in the four traits. The second PCA included the size-corrected beak traits (i.e., residuals of linear models including beak traits as the independent variable and body mass as the explanatory one). The first axis (PCA1$b$), mainly showing variability in beak shape, retained 84% of the information. Hand-wing index (HWI) was not correlated with any other trait and therefore was included independently (i.e., raw). In addition to PCA1$m$, PCA2$m$, PCA1$b$, and HWI our final dataset included trophic niche and primary lifestyle as categorical variables. Considering these six variables that were not strongly correlated (Supplementary Fig. 5), we computed a matrix of pairwise functional distances between species (Gower distance) that was used as input in a Principal Coordinates Analysis (PCoA). The relative contribution of each trait to the global Gower distances was calculated using the *kdist.cor* function in the ade4 package[52]. The contribution was relatively similar across ecological facets, being ~33% for quantitative traits defining physical characteristics (~12% PCA1$m$, ~5% PCA2$m$, ~5%, PCA1$b$, and ~12% HWI), ~33% for trophic niche, and ~33% for primary lifestyle. The matrix of distances between species was rather consistent to the removal of the two qualitative traits (Spearman correlation coefficient = 0.67, $P < 0.001$). The first five axes obtained from the PCoA accounted for ~72% of the total variation. We used five axes to define the functional space because including more axes increased explained variance poorly (6% the sixth axis and 4% the seventh axis), but incremented computational effort exponentially.

From this five-dimension functional space, we calculated the functional richness (volume of the convex hull) for each regional assemblage corresponding to each grid cell using the package *fundiversity* v.0.2.1[53]. We used functional richness *sensu stricto* in order to assess the net functional assets encompassed by a given assemblage and thereby examine the range of attributes represented in a given region[54]. We followed this approach (i.e., measure the relative contribution of each set of traits, and calculate a functional space based on PCoA dimensions) to avoid some of the issues that might arise from mixing traits of different resolutions[55].

### Land-use diversity

To assess regional land-use diversity, we calculated the Shannon index of the land-use types in each 100-kilometer grid cell. Land-use types were defined by land cover from the Copernicus Global Land Service Land Cover Map (22 classes in total; Supplementary Table 2) at a 100 m resolution (CGLS-LC100), delivered from the vegetation instrument on board of the PROBA satellite (PROBA-V)[56]. This index can theoretically range from 0, meaning that the grid is homogeneous, with no diversity (i.e., only pixels of a single land-use type present) to infinite, meaning maximum heterogeneity (i.e., each pixel of a different land-use type). We used the cloud-based platform Google Earth Engine[57] to collect and extract the land-use type information and calculate the Shannon diversity index of pixels in each bird grid. Then, the land-use diversity value for each grid cell was extracted (Fig. 1).

To explore the relationship between regional land-use diversity and other important factors that are known to potentially influence macroecological patterns of bird diversity[24,27], we also obtained for each grid cell: the median altitude (assemblages in high locations might be less rich[17,20]), the coordinates of the center (longitude and latitude)[15,16], the median annual temperature (assemblages in extremely hot or cold climates might be less rich)[23], the mean net primary productivity (from NDVI) assemblages in highly productive environments often show a higher richness[22], median human footprint index[58] (assemblages in highly anthropized environments might be less rich[26,59]) and median annual precipitation (assemblages in highly dry environments might be less rich[23,60]) (Supplementary Table 3 for details about data sources and resolutions used). Pearson correlation tests showed that some of these variables were highly correlated among each other, but only primary productivity was globally correlated with land-use diversity ($r = 0.49$, $P < 0.001$). This is expected because lower productive environments provide a more restricted set of habitat types as defined by Copernicus Land Cover Maps (Supplementary Table 2), and the proportion of area devoted to each habitat type changes by definition from poorly diverse regions to highly diverse regions (Supplementary Fig. 6). To discern between the effects of NPP and

those of land-use diversity, NPP was included as a covariate in some models (see next section).

## Statistical analyses

All the analyses were performed using R version 4.0 (R Core Team 2021).

First, to examine overall patterns of taxonomic and functional richness across biogeographic realms, we ran two similar generalized additive mixed models with log-transformed taxonomic and functional richness, respectively, as response variables. Shannon land-use diversity index with a smoothing term was the main factor of interest. To account for spatial autocorrelation and the effect of net primary productivity, the central coordinates (i.e., longitude and latitude) of grid cells and a proxy of net primary productivity (NPP measured as mean NDVI per grid) were included as covariates with smoothing terms. Smoothing was done using a basis dimension of $k = 3$ (the same for all the variables) to avoid overfitting, and a shrinkage version of the thin-plate regression spline. A random factor with realms as levels entered the model, defining random intercepts and random slopes.

Second, to study the effect of land-use diversity on regional bird taxonomic and functional richness within each biogeographic realm we fitted two generalized additive models with the log-transformed response variable taxonomic richness and functional richness respectively, being a smoothed term of Shannon land-use diversity index within each level of the factor realm (by = realm), the main explanatory variable. We also included smoothed terms of longitude and latitude of grid cells as covariates. The smoothing method used a basis dimension $k = 3$ (the same for all the variables), and a shrinkage version of the thin-plate regression spline.

Taxonomic richness is often a main driver of functional richness. Hence, to also assess the variations in functional richness not due to changes in taxonomic richness, we fitted another model for functional richness in which a smoothed term of taxonomic richness was included as a covariate. Standardized effect sizes (SES) were not used because in this scenario, they become a measure of uncertainty/variability rather than a corrected unbiased measure of richness, and therefore, its interpretation can be misleading (Supplementary Note 1 for a detailed explanation).

To disentangle the effect of land-use diversity from that of NPP alone within each realm, we fitted additional models including NPP as a covariate. To compare the relative importance of each variable, we conducted a leave-one-out jackknife procedure. Next, we compared the performance of the competing models based on the deviance explained by each of them, and their AIC. We also compared the deviance explained by each single non-redundant predictor variable. To improve result standardization and comparability, we ran linear models with the same structure (including coordinates and net primary productivity as a covariate), and calculated the partial estimated slopes of taxonomic/functional richness as response to the Shannon land-use diversity. For this, we used the *lm* function in base R.

Lastly, to gain a deeper understanding of the possible factors explaining differences across biogeographic realms, we fitted linear models using the median distribution range size of all the species in each grid cell as response variable. Despite that the distribution range of most species covers areas of unsuitable habitat, this variable has been proposed to be a proxy of niche breadth and environmental tolerance (i.e., habitat specialization), because species with narrow ranges often have a narrower Grinnellian niche compared to those species with wider distribution range sizes[45]. To analyze its possible association with land-use diversity, the Shannon land-use index was included as the focal explanatory variable interacting with realm. Also, latitude and longitude coordinates as well as net primary productivity were introduced in this model as covariates. Additive models were run using the and *mgcv v.1.8-36*[61] packages. The package *ggplot2* v.3.3.6 was used to draw plots[62], *ade4* v.1.7-20 to calculate

trait contributions[52], *lme4* v.1.1-27.1 to fit linear models[63], and *visreg* v.2.7.0 to calculate partial effects[64]. The assumption of independence and normality of model residuals was satisfied by all the models presented here. In addition, model residuals did not show spatial autocorrelation (Pearson correlation coefficients between model residuals and coordinates <0.001, *P*-values >0.95 in all cases). Finally, results were not significantly sensitive to the choice of the smoothing factor (Supplementary Fig. 7 for results when smoothing factor $k = 4$). The complete dataset we used to reproduce these analyses is publicly available (see Data availability section). Maps were drawn using the *cartopy* package v.0.21.1[65] in Phyton v. 3.10.0, base maps from Natural Earth.

## Reporting summary

Further information on research design is available in the Nature Portfolio Reporting Summary linked to this article.

## Data availability

The data generated in this study have been deposited in Figshare[66] (https://doi.org/10.6084/m9.figshare.21747257.v1). Data used to calculate functional and taxonomic bird richness are already publicly available in the AVONET dataset: (https://doi.org/10.1111/ele.13898). Bird distribution data by BirdLife International (http://datazone.birdlife.org/species/requestdis). Land-use diversity and NDVI values were calculated for each grid cell in Google Earth Engine (https://earthengine.google.com/) using land-use data from Copernicus Global Land Service website (https://proba-v-mep.esa.int/proba-v-mep-toolset/geo-viewer). Source data are provided with this paper.

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

## Acknowledgements

We thank everyone that contributed to collect and make freely available these precious datasets. We also thank Carlos P. Carmona for providing helpful discussions and insights about the functional trait analyses. CMN was supported by the "Juan de la Cierva" program (ref. FJC2021-046829-I). V. García-Navas was supported by the "Ramón y Cajal" program (ref. RYC2019-026703-I) from the Spanish Ministry of Science and Innovation. The open-access publication of this study was funded by the project COMEVO (ref. PID2021-123304NA-I00 to VGN) from the Spanish Ministry of Science and Innovation-European Social Fund.

## Author contributions

C.M.N. conceptualized the main ideas of the study, collected the datasets, performed the analyses, and wrote the first draft. R.M.P. calculated and provided the spatially explicit variables and map figures. VGN contributed key ideas for the analyses and helped to frame the study. The three authors contributed significantly to the final version of the manuscript.

## Competing interests

The authors declare no competing interests.
