## [Peer Review File · Nature Communications]

Land-use diversity predicts regional bird taxonomic and functional richness worldwideREVIEWER COMMENTS

Reviewer #1 (Remarks to the Author):

it was a pleasure to read this manuscript as the authors have done an excellent job in structuring and writing their study. The study looks into whether and how regional land use diversity influences functional and taxonomic bird richness across the globe. They accounted for potential confounding factors by including net primary productivity as covariate (and coordinates for spatial autocorrelation). They found that indeed land use diversity drives both richness metrics with slightly hump shaped curves globally (i.e. maximum at intermediate to high levels of land use diversity) and slightly variable responses per realm. The results are highly relevant for understanding biodiversity patterns at macroecological scales and in response to human-driven factors such as land use.

I don't have any major comments or critique, but some smaller aspects to consider - please see them in the attached word manuscript file which I commented in directly.

The most relevant being:

- you consider other factors driving biodiversity patterns (such as temperature, precipitation, altitude, etc.) but because only NPP correlates with land use diversity, you did not include them in the analyses. I understand this study focuses on land use diversity and NPP is added merely to account for potential confounding effects, but it would still be interesting to see how those other variables factor into the models. Especially since some of them cannot be altered by humans (e.g. altitude) whereas land use diversity is such a direct consequence of human civilization. One aspect could be to include those other factors in one combined model and calculate the relative variable importance, resulting in insights on how relevant actually land use diversity is in comparison to those other drivers

- the discussion repeatedly mentions the effects of fragmentation and potential trade-off effects between land use diversity and fragmentation (isolation) on species richness. since you used rather fine resolution land use data, measures of fragmentation would have been possible and could have been calculated at a grid level to be correlated with your richness responses too. I understand if this extends the scope of this study but you could mention this in the discussion that instead of only surmising, actually measuring the different effects of land use diversity and fragmentation would be a necessity for future studies.

- another aspect, although probably minor, is that of spatial autocorrelation. You added the coordinates as covariates to account for it, but this is not always effective. Did you test spatial AC in model residuals afterwards? If so please add this (in supplementary).

All other, more minor comments should be found in the annotated word file.

Reviewer #2 (Remarks to the Author):

Summary:

This study presents global patterns in taxonomic and functional diversity as a function of land-use diversity and other environmental variables. The authors find that land-use diversity is an important driver of both dimensions of diversity in bird communities, even when accounting for net primary productivity, climatic factors and biogeographic realm. The study is clear and well-written, although some clarifications in the terminology and functional diversity methods are needed.

Major issues:

Consistency of the key terminology: Define land-use diversity at the very beginning of the introduction as this is the key concept in your study. It's not clear from the first paragraph how land-use diversity differs from habitat heterogeneity. Often land-use refers to anthropogenic categorization of terrestrial land, while habitat refers to the type of area used by species or communities. At the beginning of the discussion you contrast land use and habitat diversities in relation to spatial scale, but don't give any more details on your definitions.

Functional diversity methods: (1.) Why do you use this particular functional diversity measure? Did you explore other measures and differences in results if you used different functional diversity measures? Argumentation should be added to the methods. One potentially useful paper: Mammola et al. (2021) DOI: 10.1111/1365-2435.13882. (2.) How were the categorical traits weighted when calculating the Gower distance matrix? Did you give each category the same weight as the continuous trait? I wonder if you have accounted for the common issue of including both categorical/binary and continuous traits into the calculation of functional diversity. That's because that is often the key criticism of using distance-based functional diversity measures. (3.) You should explain in more detail what you mean by "diet" and "primary lifestyle" traits as these could be defined in very many ways. In general, more arguments and references for the choice of traits are needed in the methods.

Novelty: As several studies have already mapped the patterns of bird community diversity (both functional and taxonomic) globally, I think the novelty of this work should be highlighted more clearly. In my view, the novelty lies in the spatial resolution (landscape scale) and understanding how land-use diversity affects bird communities, but with added focus on the anthropogenic effects on land-use that will have the strongest effects on bird communities in the future.

Scale dependency: I would have liked to see more coherent introduction and discussion on the scale dependent processes that influence different dimensions of diversity. You mention the issues multiple times, but don't really collect the pieces into clear paragraphs. Where in the spatially scalable hierarchy of biodiversity maintenance processes do you place your study aims?

Minor issues:

Line 33: Please be consistent and check the wording throughout the paper for land-use/land use

Line 42: It is not clear in the abstract how you define land use diversity versus habitat heterogeneity. Wouldn't you imagine these two are inherently linked to each other?

Line 58: Remove "too"

Line 79: Remove either "moreover" or "also"

Line 80: What does "these" refer to in this sentence?

Line 95: Rephrase "tighter" to "stronger" etc.

Line 115: Replace ", which" with "that"

Line 117: Please provide references that support your choice of traits and show the importance of the selected traits.

Line 124: Instead of "previously log-transformed", please add a sentence like "We log₁₀-transformed four traits: body mass, length of tarsus, wing length and tail length."

Line 139: Please give credit to the developers of the package by adding the citation.

Line 141: What do you mean by "this approach"?

Line 144: In some parts of the paper, you call the index Shannon-Wiener index, please be consistent throughout the text.

Line 157: Provide reasoning for including each of the additional variables, either here or in the introduction.

Line 176: Did you use the same smoothing factor for all smoothing terms? Did you test the robustness of the models by using different smoothing factors?

Line 187: You mention that taxonomic richness was the main driver of functional richness before mentioning that you included it in the model, please reorder the sentences here to clarify.

Line 203: Provide details on the fitting of the linear models (e.g. citations to R packages).

Line 209: I know range size is used as a proxy for habitat specialization, but these two properties have completely different spatial resolutions and a species' range may cover very large areas that aren't suitable at all for the species habitat-wise. I think you should acknowledge this somewhere in the paper.

Line 275-277: Again some mixing of land use and habitat terminology.

Line 295: Same as previous comment.

Line 308: Do you have references to support these four points or are they fully your ideas? Please phrase the sentence to make this distinction clear.

Discussion: I would have liked to see discussion on more concrete applications of your results. Birds are often a focal taxon for conservation purposes and it would for example be interesting how your results relate to the Important Bird Areas.

Fig 4: Add color legend for the maps on top of the graphs.

Table 1: Consider using deltaAIC instead of AIC to make it easier to interpret the model selection results here.

Supplementary table S2: Write the citations in standard format.

Reviewer #3 (Remarks to the Author):

In this manuscript the authors look at the link between land-use diversity and regional taxonomic and functional richness of birds. This is an important contribution to the literature on the effects of habitat heterogeneity and land-use diversity on species communities. The focus on land-use diversity at a regional scale can also contribute to the debate on the effects of habitat fragmentation.

Overall, the paper is well written but a few points in the methods could do with some elaboration.

Why were introduced species not included? Given that there are many examples of introduced species accumulating in anthropogenic land-use types, their exclusion here requires some justification.

Secondly, how was it decided to use five PCoA axis to create the functional space? This also links to the choice and treatment of the traits going into the Gower distance calculation. Was any testing done to determine if the results were robust to changes in these choices?

Similarly, were the results robust to the choice of land cover map classes? Some types of vegetation

structure (e.g. forest) are far more subdivided than others.

Line 145 – should “or” be “in”?

Line 170 – should this read “generalized additive mixed models”

Line 560 – it may be better to show ΔAIC relative to the chosen model instead. *[Editor's note: I would suggest showing the ΔAIC values in addition to, not instead of, the raw AIC values]*

REVIEWER COMMENTS

Reviewer #1 (Remarks to the Author):

it was a pleasure to read this manuscript as the authors have done an excellent job in structuring and writing their study. The study looks into whether and how regional land use diversity influences functional and taxonomic bird richness across the globe. They accounted for potential confounding factors by including net primary productivity as covariate (and coordinates for spatial autocorrelation). They found that indeed land use diversity drives both richness metrics with slightly hump shaped curves globally (i.e. maximum at intermediate to high levels of land use diversity) and slightly variable responses per realm. The results are highly relevant for understanding biodiversity patterns at macroecological scales and in response to human-driven factors such as land use.

I don't have any major comments or critique, but some smaller aspects to consider - please see them in the attached word manuscript file which I commented in directly.

Thank you very much for your comments and detailed review. It was really satisfying and encouraging to read such positive and constructive feedback about our work. We particularly appreciate that your comments helped us to significantly improve our study in the same direction of what we originally wanted to do.

The most relevant being:

- you consider other factors driving biodiversity patterns (such as temperature, precipitation, altitude, etc.) but because only NPP correlates with land use diversity, you did not include them in the analyses. I understand this study focuses on land use diversity and NPP is added merely to account for potential confounding effects, but it would still be interesting to see how those other variables factor into the models. Especially since some of them cannot be altered by humans (e.g. altitude) whereas land use diversity is such a direct consequence of human civilization. One aspect could be to include those other factors in one combined model and calculate the relative variable importance, resulting in insights on how relevant actually land use diversity is in comparison to those other drivers.

Thank you for this comment. We did not fit more holistic/comprehensive models because: (i) patterns of bird (functional) diversity with temperature, elevation or latitude have been explored before and some of them are already well known (e.g., Hughes et al. 2022; Quintero and Jetz 2018); and (ii): some of the explanatory variables were correlated (e.g., latitude and temperature or precipitation and NPP) (Extended Fig. 1). However, we think it is a good idea to see and show the relative contribution of the variables compared to land-use diversity. Therefore, in the new version, we include the non-redundant variable (i.e., elevation) in the leave-one-out jackknife procedure, and also report the relative contribution of each variable: coordinates (being latitude a proxy of

temperature); NPP (proxy of precipitation); land-use diversity; and elevation, to these patterns (L 364, Table 1). We also add a new extended figure (Extended Fig. 4).

- the discussion repeatedly mentions the effects of fragmentation and potential trade-off effects between land use diversity and fragmentation (isolation) on species richness. since you used rather fine resolution land use data, measures of fragmentation would have been possible and could have been calculated at a grid level to be correlated with your richness responses too. I understand if this extends the scope of this study but you could mention this in the discussion that instead of only surmising, actually measuring the different effects of land use diversity and fragmentation would be a necessity for future studies.

We wanted to calculate fragmentation values for each cell in an early stage of the study, but were not able to do so because they are computationally undoable with our current resources (and there were some conceptual problems such as the definition of the focal land-use type to which fragmentation refers). However, although we miss the important configurational aspect of fragmentation (how patches are arranged in space), it is inherent to the concept of fragmentation that land-use diversity contributes to it (e.g., there is no fragmentation if there are no different land-use types and the more land-use types the more likely fragmentation is). Nonetheless, we agree with this comment, and we mention this issue in the discussion of the new version (L 189-192).

- another aspect, although probably minor, is that of spatial autocorrelation. You added the coordinates as covariates to account for it, but this is not always effective. Did you test spatial AC in model residuals afterwards? If so please add this (in supplementary).

Thank you. We checked and provided further details about the lack of spatial autocorrelation in the residuals of all the models in the new version (L 385-387).

All other, more minor comments should be found in the annotated word file.

Thank you, we reproduce the main comments of the annotated word file here, and respond to them below each one.

L70: This seems a bit ambiguous to me. could you be more specific in what exactly you mean by landscape richness?

Yes, changed to “species richness in the landscape”.

L77: you could mention climate change here as well (although this is indirectly driven by human activities) especially with regards to desertification

We agree. We see desertification as a problem aggravated by climate change. We specify it now in the new version (L82).

L91: I suggest to remove this as it sounds a bit too strong

Done.

L116-123: this is all explained very well and clear. the only thing I'm missing is a short reasoning of why you selected those specific traits (data availability or ecological reasons)? second, were beak morphology and diet collinear as they both entered the modeling?

Thank you. We specify now in the text that we tried to include as many important facets of species' ecology (fundamental and realized Eltonian niche) as possible, to characterize their functional diversity in a comprehensive way (e.g., considering physical characteristics, main trophic niche and lifestyle), with the constrain of what was available for all the bird species in the world (L 256-259).

Additionally, we show now a table with more trait details, the ecological implications of each trait, and supporting literature (Supplementary Table 1). Any pair of traits used (including beak morphology and diet) did not show important collinearity (taking 0.7 as the commonly used threshold in these cases). We show it in Supplementary Fig. 1 of the new version).

L162: great that you included those additional variables. but apart from NPP and coordinates, the other variables were not mentioned anymore in the analyses. is it because the other variables were not correlated with land use diversity? if so, please be clearer in the next few sentences.

On the other hand, it would be interesting to see if/how the models (e.g. variance explained) improve by adding these other factors potentially driving species richness at global scale.

Interesting point. Exactly, the original scope of this study was to investigate the effect of land-use diversity, because, to our knowledge, this was not examined to date. However, we find it interesting to see the relative importance of this variable compared to others. Therefore, in the new version we include the only non-redundant variable (elevation) and show it. However, please note that other variables showed a high collinearity with some of the ones already included. For instance, temperature with latitude, or precipitation with NPP (Extended Fig. 1). (L 364-365 and Table 1).

L173: this response variable was not explained at all. It may be familiar to most readers but a very short summary of how this was calculated would be good nevertheless.

Thank you, we explain it now in the new version “The taxonomic richness of bird species in each grid cell was calculated by counting the number of different species occurring in each grid cell.” (L 251-254).

L201: but already in the first model described above you included NPP – how is this here different? Was it within each level of realm?

We apologize for the lack of clarity in this sentence. Yes, exactly, here, we look at the effects within each realm. We first ran an overall model (with realms as random, and NPP as covariate), then in a second step, we looked at trends within realms, and we fitted models with and without NPP as covariate (to disentangle what part of the effects within each realm is due to NPP and what is due to land-use diversity alone). We hope it is more clearly explained now in the new version (L 346, L 360-362).

L204: this basically describes a leave-one-out jackknife procedure to calculate rel. variable importance. maybe it would be easier if you just said this instead of listing the different models?

True! We describe it as suggested in the new version (L 363).

L252: of taxonomic or functional or both?

Thank you. Particularly in functional diversity, we clarify it in the new version (L 126).

L282: since you bring this up here, I suggest it’s a good location to add your findings on the NPP influence on f. and t. richness.

Thank you, done (L 160-163).

L309: also known to be highly scale-specific (i.e. variable effects depending on the spatial scale).

Good point. We include it now (L 188-189).

L337: please add a reference here

Done, we added 4 new references to support our idea (L 218- 221).

Reviewer #2 (Remarks to the Author):

Summary:

This study presents global patterns in taxonomic and functional diversity as a function of land-use diversity and other environmental variables. The authors find that land-use diversity is an important driver of both dimensions of diversity in bird communities, even when accounting for net primary productivity, climatic factors and biogeographic realm. The study is clear and well-written, although some clarifications in the terminology and functional diversity methods are needed.

Thank you very much for the time dedicated to review our study, and for the interesting and thoughtful comments provided. They helped us to improve the quality, clarity and reproducibility of our study.

Major issues:

Consistency of the key terminology: Define land-use diversity at the very beginning of the introduction as this is the key concept in your study. It's not clear from the first paragraph how land-use diversity differs from habitat heterogeneity. Often land-use refers to anthropogenic categorization of terrestrial land, while habitat refers to the type of area used by species or communities. At the beginning of the discussion you contrast land use and habitat diversities in relation to spatial scale, but don't give any more details on your definitions.

Thank you, we apologize for this missing definition. We have clarified the terminology in the new version (L 34, L43, L 55-L56, L 62, L 69).

Functional diversity methods:

(1.) Why do you use this particular functional diversity measure? Did you explore other measures and differences in results if you used different functional diversity measures? Argumentation should be added to the methods. One potentially useful paper: Mammola et al. (2021) DOI: 10.1111/1365-2435.13882.

We used this functional measure because it represents well the net functional "assets" encompassed by a given species assemblage. Other commonly used measures, such as the Rao Entropy/functional divergence, functional dispersion or functional evenness, focus more on aspects such as functional differences or functional redundancy among species. Some of these other measures can even show increases when the species set is reduced, and are often used for example to examine the level of niche differentiation. We did not want to focus on this. We pay attention to functional diversity *sensu stricto* in order to examine the range of attributes represented in a given location. In the new version, we justify our choice for this functional diversity measure and cite the suggested paper (L 293-295).

We only calculated this functional measure because it is computationally very expensive and time-consuming to calculate multiple functional diversity measures for such a large number of species assemblages, many of them including hundreds of species.

(2.) How were the categorical traits weighted when calculating the Gower distance matrix? Did you give each category the same weight as the continuous trait? I wonder if you have accounted for the common issue of including both categorical/binary and continuous traits into the calculation of functional diversity. That's because that is often the key criticism of using distance-based functional diversity measures.

Thank you for raising this important point. We apologize about this information being missing in the previous version. In the new version we show the relative contribution of each trait to the gower distances (~33% quantitative variables defining physical characteristics, ~33% trophic niche and ~33% primary lifestyle). We are aware about the potential issues of mixing variables of different resolution. However, we are confident that this is not an issue in our study (as long as we know and show their relative contribution) because: (i) the main problem with using categorical variables to define functional spaces is that they “can mask functional variability and inflate functional redundancy among species” (Kohli and Jarzyna 2021), especially when they have a low number of levels (e.g., binary traits). Then, our analyses are likely estimating the importance of land-use diversity on functional richness conservatively; and ii) we trust that the functional space is better defined if we include specific information even if it is not available as continuous. Otherwise, we would lose important information to satisfy data resolution homogeneity. In other words, we would bias the use of available information to avoid bias in the relative importance of this information. We show how the functional space is defined in terms of weights in the new version and report that a distance matrix based only on the quantitative variables is rather similar to the one calculated in this study (Spearman correlation coefficient = 0.67, $P < 0.001$) (L 280-286).

(3.) You should explain in more detail what you mean by "diet" and "primary lifestyle" traits as these could be defined in very many ways. In general, more arguments and references for the choice of traits are needed in the methods.

In the new version, we clarified the term diet “predominant trophic niche”. In addition, we show a new table (Table S1), in which we provide more details about all the pre-selected traits, and their ecological importance (with supporting literature).

Novelty: As several studies have already mapped the patterns of bird community diversity (both functional and taxonomic) globally, I think the novelty of this work should be highlighted more clearly. In my view, the novelty lies in the spatial resolution (landscape scale) and understanding how land-use diversity affects bird communities, but with added focus on the anthropogenic

effects on land-use that will have the strongest effects on bird communities in the future.

Thank you for this comment. We have emphasized the novelty in the abstract and the introduction with sentences such as: “the role of land-use diversity in structuring global taxonomic and functional richness is unknown”, and: “Here, we examine the hypothesis that regional species taxonomic and functional richness is driven by global patterns of land-use diversity”. And, in the introduction: “Yet, a universal direct link between land-use diversity and biodiversity patterns is still missing”. We make explicit reference to these previous studies, and indicate that our study fills a gap and thus, constitute a further step towards a better understanding of factors shaping diversity of bird assemblages at global scale.

Scale dependency: I would have liked to see more coherent introduction and discussion on the scale dependent processes that influence different dimensions of diversity. You mention the issues multiple times, but don't really collect the pieces into clear paragraphs. Where in the spatially scalable hierarchy of biodiversity maintenance processes do you place your study aims?

We place it at the regional scale because it is the only scale at which we have bird information (grid cell of ~100 km² is the sampling unit). The regional scale is explicitly mentioned from the title throughout the manuscript. In the new version we introduced clear definitions of land-use diversity (measured regional as Shannon index) and habitat or land-use quality (measured as regional NPP; NDVI) because the lack of these definitions probably contributed to unclarity in the previous version (L 55-56; L -69-70).

Minor issues:

Line 33: Please be consistent and check the wording throughout the paper for land-use/land use

Done, thank you!

Line 42: It is not clear in the abstract how you define land use diversity versus habitat heterogeneity. Wouldn't you imagine these two are inherently linked to each other?

We refer to habitat heterogeneity (measured as NPP) as to habitat quality or resource availability (e.g., tropical forests have high habitat heterogeneity, and deserts, very low), and land-use diversity as to variability of land-use types. We make it clear now in the new version to avoid confusion (L 34 and L 43-44).

Line 58: Remove "too"

Done, thank you.

Line 79: Remove either "moreover" or "also"

Done.

Line 80: What does "these" refer to in this sentence?

We rephrased the sentence to make it clearer.

Line 95: Rephrase "tighter" to "stronger" etc.

Done, thank you.

Line 115: Replace ", which" with "that"

Done.

Line 117: Please provide references that support your choice of traits and show the importance of the selected traits.

Thank you. In the updated version, we added a new table (Supplementary Table 1) where we discuss the ecological significance of the features employed and provide references to back up these claims.

Line 124: Instead of "previously log-transformed", please add a sentence like "We log₁₀-transformed four traits: body mass, length of tarsus, wing length and tail length."

Done.

Line 139: Please give credit to the developers of the package by adding the citation.

Done. We apologize for this missing reference.

Line 141: What do you mean by "this approach"?

We clarify it in the new version.

Line 144: In some parts of the paper, you call the index Shannon-Wiener index, please be consistent throughout the text.

Thank you, done in the new version.

Line 157: Provide reasoning for including each of the additional variables, either here or in the introduction.

Done. We provide additional reasoning about the consideration of these specific variables (L 314-322).

Line 176: Did you use the same smoothing factor for all smoothing terms? Did you test the robustness of the models by using different smoothing factors?

Yes, we specify this in the new version. In addition, we show the (low) sensitivity of results to a higher smoothing factor ($k = 4$) in Supplementary Fig.3. Please, note that we used a smoothing factor of $k = 3$ because it avoids overfitting in models aiming to find patterns in such big datasets with significant noise. Bigger smoothing factors are not recommended because they are very sensitive to noise (increased “wiggleness” lowers the capacity of ecological interpretation), and can render potentially spurious results (especially in the extremes of the x variable/axis because there is less data when the explanatory variable is normally distributed).

Line 187: You mention that taxonomic richness was the main driver of functional richness before mentioning that you included it in the model, please reorder the sentences here to clarify.

Done, thank you.

Line 203: Provide details on the fitting of the linear models (e.g. citations to R packages).

Done.

Line 209: I know range size is used as a proxy for habitat specialization, but these two properties have completely different spatial resolutions and a species' range may cover very large areas that aren't suitable at all for the species habitat-wise. I think you should acknowledge this somewhere in the paper.

We fully agree and acknowledge this in the new version (L 374-375). We used it because it is the only proxy of habitat specialization available for all the bird species in the world (that we are aware of). In addition, please, note that this difference has no strong implications for our analyses and conclusions, because

range size still indicates how likely a species is to occur in different land-use or habitat types (Slatyer, Hirst, and Sexton 2013).

Line 275-277: Again some mixing of land use and habitat terminology.

We reworded it in some sites throughout the text. However, please, note that there is a strong link between these two terms, since different land-use types represent also different habitat types (although strictly, "habitat" should refer to a individual specific species). We hope that defining the two aspects (habitat quality/heterogeneity = NPP vs. land-use diversity from the abstract and introduction make it clearer now).

Line 295: Same as previous comment.

Rephrased.

Line 308: Do you have references to support these four points or are they fully your ideas? Please phrase the sentence to make this distinction clear.

We originally present them as own ideas, but we now use two references to support the second point.

Discussion: I would have liked to see discussion on more concrete applications of your results. Birds are often a focal taxon for conservation purposes and it would for example be interesting how your results relate to the Important Bird Areas.

Thank you for this suggestion. We mention the more applied implication suggested by the reviewer in the new version (L 226-229). However, please note that our aim is to show this general pattern that can then be applied to countless situations by others (e.g., assess the suitability or efficiency of Important Bird Areas). We think that this could be a very interesting whole topic for another study.

Fig 4: Add color legend for the maps on top of the graphs.

Done, thank you.

Table 1: Consider using deltaAIC instead of AIC to make it easier to interpret the model selection results here.

Thank you, we included deltaAIC in addition to AIC in the new version.

Supplementary table S2: Write the citations in standard format.

Done, thank you.

Reviewer #3 (Remarks to the Author):

In this manuscript the authors look at the link between land-use diversity and regional taxonomic and functional richness of birds. This is an important contribution to the literature on the effects of habitat heterogeneity and land-use diversity on species communities. The focus on land-use diversity at a regional scale can also contribute to the debate on the effects of habitat fragmentation. Overall, the paper is well written but a few points in the methods could do with some elaboration.

Thank you for your feedback. We sincerely appreciate your time and comments. We feel it's crucial to clarify the points you raised.

Why were introduced species not included? Given that there are many examples of introduced species accumulating in anthropogenic land-use types, their exclusion here requires some justification.

Thank you for helping us to clarify this point. We did not include Introduced species as a conservative measure because they could introduce some bias (some regions are more prone to host alien species due to different factors including tourism and commercial transactions) and add noise to the results (e.g., differences in functional diversity could be disproportionately inflated in some areas compared to others due to particular (stochastic) historical events in form of species introductions. We were interested in understanding a fundamental pattern in species assemblages, so there should be a strong ecological signal even if the process of species introductions was not included. However, we decided to leave this process out of the game to be more conservative and avoid the criticism that this fundamental pattern could be driven by introduced species. In the new version, we justify in more detail the exclusion of introduced species (238-242).

Secondly, how was it decided to use five PCoA axis to create the functional space? This also links to the choice and treatment of the traits going into the Gower distance calculation. Was any testing done to determine if the results were robust to changes in these choices?

We apologize for this missing information. We decided to use five axes because there was relatively little gain on including more axes in terms of explained variance for the high cost of making analyses exponentially more expensive. In

the new version of the manuscript, we justify this decision: “We used these five axes to define the functional space because including more axes increased explained variance poorly (6% the sixth axis and 4% the seventh axis), but incremented computational effort exponentially”. Please, note that using five axes is a relatively high number; most studies use 3-7 axes at most despite being conducted on a much smaller scale (tens or hundreds of species instead of thousands) (L 288-291).

We did not run analyses using a different set of traits because a worldwide analysis using this many traits is computationally very hard to perform, but we show that the Gower distance matrix among species is rather consistent even when removing the two qualitative variables (Spearman correlation coefficient = 0.67, $P < 0.001$). Please, note that this is the first study to our knowledge, working with this many species traits and assemblages, so it has an unprecedented resolution. (L 286-287).

Similarly, were the results robust to the choice of land cover map classes? Some types of vegetation structure (e.g. forest) are far more subdivided than others.

Thank you for raising this interesting point. This is revealed by the fact that land-use diversity partially correlates with net primary productivity, and a reason why we included the latter as a covariate. We are convinced that our approach is robust and coherent because: (i) we already control for this effect in the models that include NPP as a covariate, which penalizes the forest subdivision; (ii) different types of forests can behave as different habitats for bird species (e.g., tree density, or the presence/absence of a shrub layer can be very important); and (iii) we feel that it is probably more robust to keep the official land-use types described by an independent source such as Copernicus Global Land Service, than making our own (probably more subjective) classification. In the new version, we emphasized that we included NPP as a covariate to disentangle the effects of NPP and land-use diversity (L 330-331).

Line 145 – should “or” be “in”?

Yes, thank you.

Line 170 – should this read “generalized additive mixed models”

Yes, thank you.

Line 560 – it may be better to show ΔAIC relative to the chosen model instead. *[Editor's note: I would suggest showing the ΔAIC values in addition to, not instead of, the raw AIC values]*

Thank you. In the new version we included the deltaAIC in addition to AIC.

References

- Hughes, Emma C., David P. Edwards, Jen A. Bright, Elliot J. R. Capp, Christopher R. Cooney, Zoë K. Varley, and Gavin H. Thomas. 2022. "Global Biogeographic Patterns of Avian Morphological Diversity." *Ecology Letters* 25(3):598–610. doi: 10.1111/ELE.13905.
- Kohli, Brooks A., and Marta A. Jarzyna. 2021. "Pitfalls of Ignoring Trait Resolution When Drawing Conclusions about Ecological Processes." *Global Ecology and Biogeography* 30(5):1139–52. doi: 10.1111/GEB.13275.
- Quintero, Ignacio, and Walter Jetz. 2018. "Global Elevational Diversity and Diversification of Birds." *Nature* 2018 555:7695 555(7695):246–50. doi: 10.1038/nature25794.
- Slatyer, Rachel A., Megan Hirst, and Jason P. Sexton. 2013. "Niche Breadth Predicts Geographical Range Size: A General Ecological Pattern." *Ecology Letters* 16(8):1104–14. doi: 10.1111/ELE.12140.

REVIEWERS' COMMENTS

Reviewer #1 (Remarks to the Author):

The authors did an excellent job reviewing their manuscript based on the comments and suggestions from the 3 reviewers. In my opinion they have addressed all comments in a satisfactory way and improved/edited the ms where necessary.

Especially the methods are much clearer now and I appreciate the addition of some more models and figures, further explaining the used methods and deepening the findings from this interesting study. I have only added a few very minor edits in the word file directly.

Reviewer #2 (Remarks to the Author):

The authors have done a great job at addressing my (Reviewer #2) and other reviewers' comments. I particularly appreciate the more detailed method description. I have no further suggestions.

Reviewer #3 (Remarks to the Author):

The concerns raised in my previous review have now been adequately addressed in the revised manuscript.

Please see below a few minor text changes -

Lines 34, 55 and 146 – land use should be hyphenated for consistency

Line 82 – perhaps “aggravated by ongoing climate change” would sound better

Lines 120 to 121 – missing superscript on R squared

Supplementary Table 1 – typo on Insectorial and citation format error Fitzpatrick S (1987)

REVIEWER COMMENTS (our responses in blue)

Reviewer #1:

The authors did an excellent job reviewing their manuscript based on the comments and suggestions from the 3 reviewers. In my opinion they have addressed all comments in a satisfactory way and improved/edited the ms where necessary. Especially the methods are much clearer now and I appreciate the addition of some more models and figures, further explaining the used methods and deepening the findings from this interesting study.

I have only added a few very minor edits in the word file directly.

Thank you for the suggested edits, we implemented all of them.

Reviewer #2:

The authors have done a great job at addressing my (Reviewer #2) and other reviewers' comments. I particularly appreciate the more detailed method description. I have no further suggestions.

Thank you very much for your review and your comments.

Reviewer #3:

Please see below a few minor text changes -

Lines 34, 55 and 146 – land use should be hyphenated for consistency

Line 82 – perhaps “aggravated by ongoing climate change” would sound better

Lines 120 to 121 – missing superscript on R squared

Supplementary Table 1 – typo on Insessorial and citation format error Fitzpatrick S (1987)

Thank you for your new suggestions, we implemented all of them.